# *WorldSense*: Evaluating Real-world Omnimodal Understanding for Multimodal LLMs

**Jack Hong[1], Shilin Yan[1][†], Jiayin Cai[1], Xiaolong Jiang[1], Yao Hu[1], Weidi Xie[2][‡]**

[1]Xiaohongshu Inc.    [2]Shanghai Jiao Tong University

{jaaackhong, tattoo.ysl}@gmail.com   weidi@sjtu.edu.cn

Project Page: https://jaaackhongggg.github.io/WorldSense

## Abstract

We introduce *WorldSense*, the *first* benchmark to assess the multi-modal video understanding, that simultaneously encompasses *visual, audio, and text* inputs. In contrast to existing benchmarks, our *WorldSense* has several features: (i) **collaboration of omni-modality**, we design the evaluation tasks to feature a strong coupling of audio and video, requiring models to effectively utilize the synergistic perception of omni-modality; (ii) **diversity of videos and tasks**, *WorldSense* encompasses a diverse collection of 1,662 audio-visual synchronised videos, systematically categorized into 8 primary domains and 67 fine-grained subcategories to cover the broad scenarios, and 3,172 multi-choice QA pairs across 26 distinct tasks to enable the comprehensive evaluation; (iii) **high-quality annotations**, all the QA pairs are manually labeled by 80 expert annotators with multiple rounds of correction to ensure quality. Based on our *WorldSense*, we extensively evaluate various state-of-the-art models. The experimental results indicate that existing models face significant challenges in understanding real-world scenarios (65.1% best accuracy). By analyzing the limitations of current models, we aim to provide valuable insight to guide development of real-world understanding. We hope our *WorldSense* can provide a platform for evaluating the ability in constructing and understanding coherent contexts from omni-modality.

## 1 Introduction

The ability to comprehend and reason about multimodal inputs—ranging from visual and textual to auditory, tactile, and beyond—is fundamental for both human and artificial agents to navigate and interpret the world. For example, when driving a car, a human driver integrates visual information (*e.g.*, recognizing road signs, traffic lights, and obstacles), auditory cues (*e.g.*, hearing the honking of another car or a siren approaching from behind), and tactile feedback (*e.g.*, the feel of the steering wheel, the vibrations of the road, or the responsiveness of the brakes) to make real-time decisions and ensure safe navigation. This seamless multimodal integration enables intelligent agents to process complex, dynamic environments and respond to subtle cues—an ability that is essential for both human perception and development of embodied agents designed to interact naturally in the world.

In the recent literature, the development of Multi-modal Large Language Models (MLLMs) (OpenAI, 2023; Hurst et al., 2024; OpenAI; Team et al., 2023; 2024b; Zhang et al., 2023; Ma et al., 2024; Fang et al., 2023) have led to remarkable progress on a series of tasks, for example, classification (Liu et al., 2024c), captioning (Alayrac et al., 2022; Dai et al., 2023; Liu et al., 2024b), question-answering (Tang et al., 2024; Panagopoulou et al., 2023; Liu et al., 2024f), OCR (Mathew et al., 2021; Zhang et al., 2024b), segmentation (Lai et al., 2024; Xia et al., 2024; He et al., 2024a), autonomous driving (Nie et al., 2025; Sima et al., 2025; Chen et al., 2024a) and more. However, multi-modal analysis primarily focuses on visual-language information, leaving out crucial modalities like audio, which results in an incomplete evaluation of their multimodal capabilities. While

---

† Project leader.   ‡ Corresponding author.

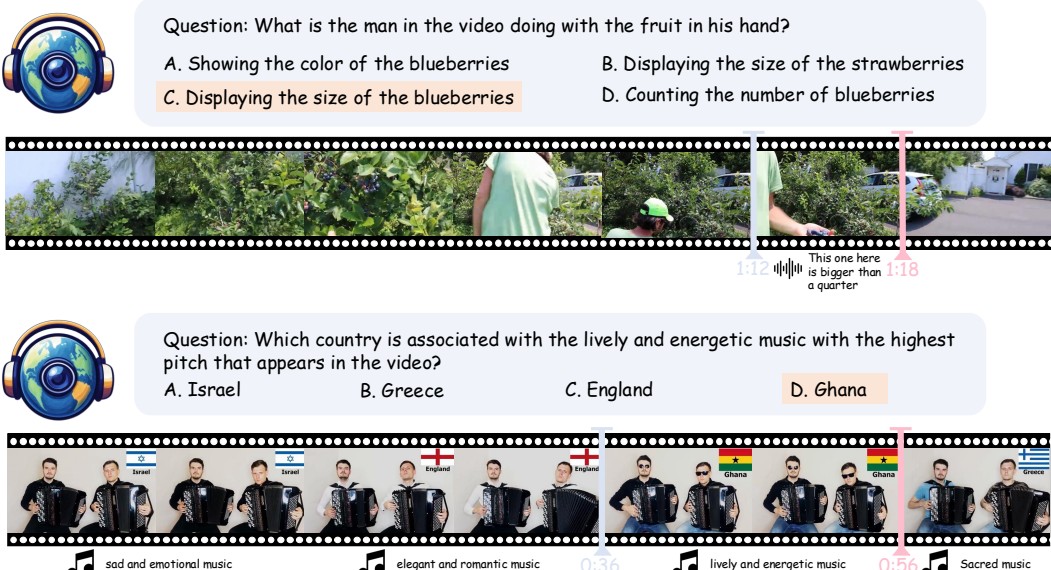

Figure 1: **Examples in *WorldSense*.** *WorldSense* highlights the importance of tightly coupled audio-visual perception for real-world understanding, where neither modality alone provides sufficient context for correct answer. In the **first** example, the video shows a man holding a fruit. However, visual information alone reveals the object, and only audio clarifies the action. In the **second** example, identifying cultural elements and locating the "lively and energetic" music segment requires both visual and auditory cues. *WorldSense* offers a platform to evaluate MLLMs' real-world perception and omni-modal understanding capabilities.

some benchmarks have started incorporating both visual and audio modalities, they still exhibit several limitations. For example, OmniBench (Li et al., 2024d) and AV-Odyssey Bench (Gong et al., 2024) mainly emphasize image evaluation, whereas other benchmarks (Geng et al., 2024; Li et al., 2022; Yang et al., 2022) either restrict to captioning tasks or are limited to simple scenarios, or suffer from low-quality, monotonous questioning patterns.

This paper presents *WorldSense*, the first comprehensive benchmark designed to evaluate Multimodal Large Language Models (MLLMs) in perceiving, understanding, and reasoning with omni-modal information in real-world settings. The benchmark is defined by three key features: **(i) Omni-modal integration.** The benchmark emphasizes the joint processing of audio and visual modalities, as illustrated in Figure 1. Each question requires both modalities for accurate response—removing either results in failure—enabling rigorous assessment of a model's capacity for integrated sensory understanding. **(ii) Diverse videos and task coverage.** The benchmark includes 1,662 synchronized audio-visual videos spanning 8 domains and 67 fine-grained subcategories. It features 3,172 multiple-choice questions across 26 cognitive tasks, ranging from basic perception to high-level reasoning. This diversity supports systematic evaluation of multimodal comprehension across a broad task spectrum. **(iii) High-quality annotations.** All question-answer pairs are curated by 80 expert annotators and undergo multiple validation rounds, including human review and automated MLLM verification. This ensures annotation accuracy and benchmark reliability. Through these methodological advancements, *WorldSense* sets a new standard for evaluating MLLMs in real-world multimodal reasoning, advancing the field toward more human-like understanding.

We conduct extensive evaluations for a broad spectrum of MLLMs, including open-source video models, video-audio models, and proprietary systems. Results reveal significant limitations in current models' ability to reason over omni-modal inputs in real-world contexts. Specifically, open-source video-audio models, despite processing both modalities, achieve only 25% accuracy—comparable to random guessing. In contrast, proprietary models such as Gemini 2.5 Pro reach up to 65.1% accuracy. However, when restricted to a single modality (audio or video), existing model's performance drops greatly, highlighting the critical role of integrated modality processing.

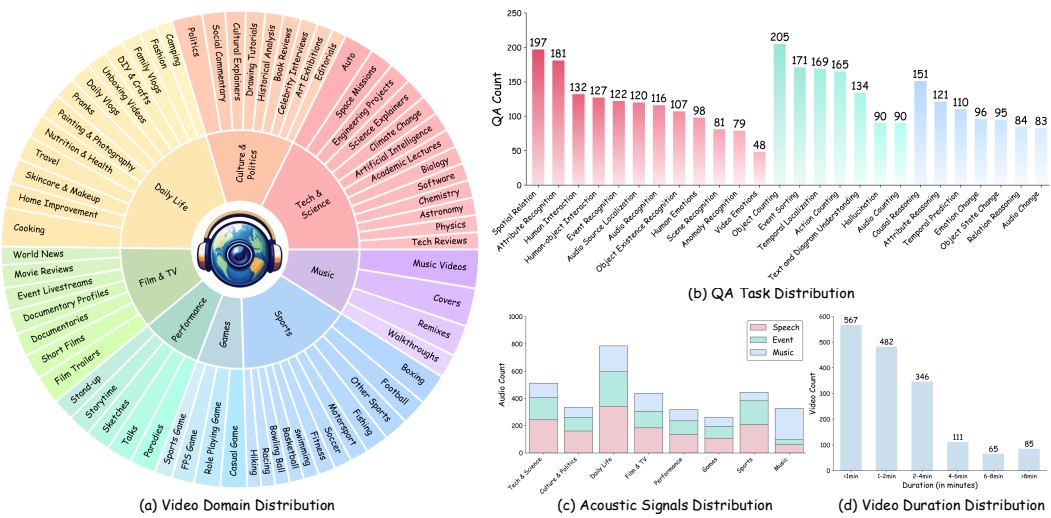

Figure 2: **Distribution of *WorldSense*.** (a) Videos in *WorldSense* spans 8 primary categories with 67 fine-grained subcategories. (b) QA pairs are structured across 26 tasks. (c) Acoustic signals distribution. Individual videos may contain multiple audio categories, leading to overlapping counts in statistical analysis. Consequently, the cumulative sum of audio instances exceeds the total video count. (d) Video duration distribution. The average duration of videos is 141.1 seconds.

We further conduct ablation studies to dissect modality contributions. Visual inputs are essential, while audio—especially raw signals—yields additional gains over text transcriptions, due to pre-served paralinguistic cues, *e.g.*, prosody, intonation, acoustic context. These findings affirm the complementary nature of audio-visual information and the necessity of their joint modeling for robust real-world understanding. Failure case analysis reveals persistent limitations in current MLLMs, motivating future directions for improving multimodal reasoning.

To summarize, we have made the following contributions: (i) we present *WorldSense*, the ***first*** benchmark tailored for evaluating MLLMs' ability on omni-modal video understanding, characterized by integrated audio-visual inputs, diverse content, and high-quality question-answering annotations; (ii) we have conducted extensive evaluation of existing MLLMs, showing that most open-source models perform near chance, and even the best proprietary model achieves only 65% accuracy—exposing a significant gap in real-world omni-modal reasoning; (iii) through ablation and failure analysis, we identify the key factors influencing performance, including raw audio and visual cues, and provide actionable insights to guide future omni-modal understanding design.

## 2 RELATED WORK

**Multimodal Large Language Models.** Current Large Language Models (LLMs) are capable of processing multimodal information, including visual, text, and audio. Early works, such as (Zhang et al., 2023; Liu et al., 2024b; Zhu et al., 2023; Driess et al., 2023; Wang et al., 2024d; Pi et al., 2023), successfully combine vision and text modalities. Subsequent research extends to temporal understanding (Wang et al., 2024f; Hurst et al., 2024; Team et al., 2024a; Liu et al., 2024e; Wang et al., 2024b;e; Li et al., 2024a; Fang et al., 2024b; Xu et al., 2024; Zhang et al., 2024a; Tong et al., 2024; Chen et al., 2024c; Lu et al., 2024a; Liu et al., 2024a), while parallel efforts (Tang et al., 2023; Chu et al., 2023; 2024) focus on audio processing. Recently, researchers shift attention to models (Cheng et al., 2024; Sun et al., 2024; Team et al., 2024a; Lu et al., 2024b; Team et al., 2024b) capable of simultaneously processing text, vision, and audio inputs. Despite the growing interest in the models which can perform the omnimodality understanding, the absence of a comprehensive evaluation benchmark restricts the development. To address this limitation, we introduce our *WorldSense* to evaluate models' capabilities in perceiving and understanding real world omnimodal scenarios.

**Multimodal Benchmarks.** The development of MLLMs has been driven by benchmarks, evolving from static image understanding (Zhang et al., 2024c; Liu et al., 2025; Li et al., 2023; 2024b; Fu

et al., 2024a; Yue et al., 2024) to temporal comprehension (Li et al., 2024c; Liu et al., 2024d; Song et al., 2024; Zhou et al., 2024; Fang et al., 2024a; Fu et al., 2024b; He et al., 2024b; Wang et al., 2024c; Xu et al., 2017; Yu et al., 2019; Lin et al., 2024; Chandrasegaran et al., 2024). However, these benchmarks largely overlook the crucial role of audio in real-world perception. While several audio-visual benchmarks have been proposed, they face significant limitations. AV-Odyssey Bench (Gong et al., 2024) and OmniBench (Li et al., 2024d) focus on static images, Music-AVQA (Li et al., 2022) and AVQA (Yang et al., 2022) are domain-specific with monotonous questions, and Long-VALE (Geng et al., 2024) limits its assessment to captioning capabilities alone. Given that existing benchmarks fail to provide a comprehensive evaluation of MLLMs' real-world understanding capabilities, we introduce *WorldSense* to address this critical gap in the field.

# 3 *WorldSense*

In this section, we first introduce the design principles in Section 3.1, followed by a description of the data collection (Section 3.2) and annotation processes ( 3.3). We then compare statics of *WorldSense* with previous benchmarks in Section 3.4, and finally present our evaluation methodology 3.5.

## 3.1 DESIGN PRINCIPLE

As for multi-modal evaluation, we base on the audio-visual synchronized videos, which capture temporal events, motion patterns, and audio-visual correlations. To curate the benchmark, we adhere to the following three principles, to ensure rigorous and comprehensive evaluations for MLLMs.

**Comprehensive Domain Coverage.** To capture the diversity of real-world scenarios, we construct a hierarchical taxonomy starting from broad human-centric domains, refined into 67 fine-grained subcategories. This structure ensures wide ecological coverage, enabling robust assessment of multimodal understanding across varied contexts.

**Diverse Acoustic Modalities.** Real-world audio can be broadly classified into speech, environmental events, and music. The benchmark includes all three types, enabling evaluation across a spectrum of acoustic complexity—from linguistic content to non-verbal and abstract auditory cues.

**Multilevel Cognitive Assessment.** We design a three-tiered evaluation framework targeting: **recognition** (detection of basic audio-visual elements), **understanding** (comprehension of multimodal relationships), and **reasoning** (high-level inference tasks such as causal inference or abstract thinking). The benchmark includes 26 tasks aligned with these levels, encouraging holistic evaluation of perceptual and cognitive capabilities in multimodal settings.

## 3.2 DATA COLLECTION & CURATION

We primarily source our video content from FineVideo (Farré et al., 2024), a large-scale dataset comprising high-quality YouTube videos that exhibit strong audio-visual correlations across diverse real-world scenarios. To enrich the benchmark's coverage of musical content, we supplement it with selected videos from MusicAVQA (Li et al., 2022), ensuring a more balanced representation of auditory modalities.

Our data collection employs a systematic filtering pipeline to ensure high-quality videos with rich visual-audio semantics and temporal dynamics, following three main steps in Figure 3(a): (i) filtering videos according to predefined taxonomic categories delineated in Section 3.1; (ii) selecting clips based on pre-computed audio-visual correlation and dynamic content metrics from about 8,000 initial videos; and (iii) human expert review for video quality and real-world relevance. This rigorous selection and processing results in 1,662 high-quality video segments with strong audio-visual correlations across various real-world scenarios.

## 3.3 ANNOTATION PROTOCOL

**Question-Answering (QA) Annotation.** A team of 80 professional annotators is engaged in creating high-quality multiple-choice QA pairs for each video by thoroughly reviewing both visual and audio content. The questions are designed to require integration of multiple modalities, enabling effective assessment of MLLMs' multimodal understanding.

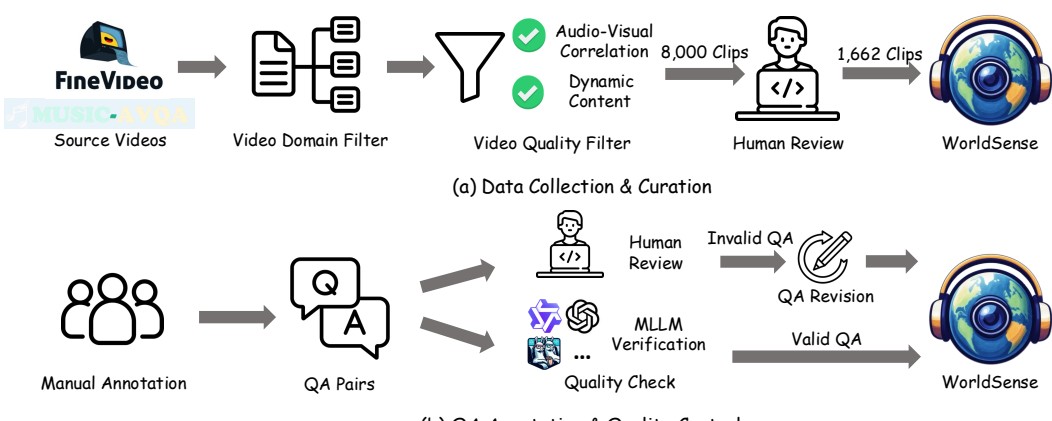

Figure 3: **Data collection and QA annotation pipelines.** (a) Data collection and curation process. (b) QA annotation and quality control pipeline.

**Quality Control.** To ensure QA quality, we implement a rigorous quality control process combining expert review and automated checks, as illustrated in Figure 3(b). Professional quality control experts evaluate each QA pair based on three essential criteria: (i) linguistic clarity and coherence, (ii) multimodal necessity for correct answers, and (iii) appropriate difficulty. Questions that fail to meet these standards are returned for revision.

We also use MLLMs for automated verification. Vision-language models like Qwen2-VL(Wang et al., 2024b) verify that questions require multiple modalities for correct answers. Furthermore, multimodal MLLMs capable of processing video, audio, and text, such as Video-LLaMA2(Cheng et al., 2024) and OneLLM (Han et al., 2024) are used to assess question difficulty, with questions answered correctly by all models being flagged for manual revision as too simple.

This dual-verification system, combining expert review and automated testing, ensures that all questions in our benchmark are of high-quality and well-formulated, that requires multi-modal comprehension, and present significant challenges for the models.

### 3.4 DATASET STATISTICS

As summarized in Table 1, our proposed *WorldSense* benchmark contains 1,662 video clips with synchronized audio across 8 categories and 67 subcategories, averaging 141.1 seconds in length, including 3,173 multiple-choice questions on three cognitive levels.

*WorldSense* features diverse audio types such as speech, environmental sounds, and music. Unlike existing benchmarks that use static images (*e.g.*, AV-Odyssey Bench (Gong et al., 2024), OmniBench (Li et al., 2024d)) or feature weak audio-visual correlations (*e.g.*, Video-MME (Fu et al., 2024b)), *WorldSense* is the first to comprehensively evaluate MLLMs' real-world multimodal understanding. It distinguishes itself through: (i) open-domain videos with multi-task evaluation, (ii) original audio-visual content with complete transcriptions, and (iii) carefully crafted questions requiring true audio-visual integration, establishing a comprehensive benchmark for real-world multimodal understanding assessment.

### 3.5 EVALUATION PARADIGM

In our evaluation framework, each test instance consists of a video clip with synchronized audio and a multiple-choice question. Models must process these multi-modal inputs and select the correct answer from several options. Performance is measured by accuracy, comparing the model's selection to the ground-truth answers. A model's success is determined by its ability to accurately align with the correct answer. We employ a matching-based approach to extract answers.

To rigorously assess the necessity of multimodal integration in real-world understanding, we conduct ablation studies across various modality configurations. This approach not only evaluates overall

Table 1: **Statistics.** A, V, I for modality represent audio, video, and image. **Len.** refers to the mean video duration in seconds. A and M for **Anno.** indicate automatic and manual annotation generation. **QA Tokens** represents the average token count in QA pairs, while **Sub. Tokens** denotes the mean number of subtitle tokens. **Multi-task** represents whether the dataset encompasses more than two question categories. **Open-domain** signifies whether the video content spans diverse domains. **Sub./Aud.** ispecifies the availability of audio signals or subtitle transcriptions. **A-V Correlations** indicates whether answering questions requires integration of omnimodal information.

| Benchmarks | Modality | #Videos | Len.(s) | #QA Pairs | Anno. | QA Tokens | Sub. Tokens | Multi task | Open domain | Sub./ Aud. | A-V Correlations |
|---|---|---|---|---|---|---|---|---|---|---|---|
| MSRVTT-QA (Xu et al., 2017) | V | 2,990 | 15.2 | 72,821 | A | 8.4 | ✗ | ✗ | ✓ | ✗ | ✗ |
| ActivityNet-QA (Yu et al., 2019) | V | 800 | 111.4 | 8,000 | M | 10.2 | ✗ | ✗ | ✗ | ✗ | ✗ |
| MVBench (Li et al., 2024c) | V | 3,641 | 16.0 | 4,000 | A | 27.3 | ✗ | ✓ | ✓ | ✗ | ✗ |
| MovieChat (Song et al., 2024) | V | 130 | 500.0 | 1,950 | M | - | ✗ | ✗ | ✓ | ✗ | ✗ |
| Video-Bench (Ning et al., 2023) | V | 5,917 | 56.0 | 17,036 | A&M | 21.3 | ✗ | ✓ | ✓ | ✗ | ✗ |
| EgoSchema (Mangalam et al., 2023) | V | 5,063 | 180.0 | 5,063 | A&M | 126.8 | ✗ | ✓ | ✗ | ✗ | ✗ |
| Video-MME (Fu et al., 2024b) | V | 900 | 1017.9 | 2,700 | M | 35.7 | 3086.5 | ✓ | ✓ | ✓ | ✗ |
| MMBench-Video (Fang et al., 2024a) | V | 609 | 165.4 | 1,998 | M | 19.3 | ✗ | ✓ | ✓ | ✗ | ✗ |
| AVQA (Yang et al., 2022) | A+V | 57,000 | 10 | 57,335 | M | 14.2 | ✗ | ✗ | ✓ | ✓ | ✓ |
| Music-AVQA (Li et al., 2022) | A+V | 9,288 | 60 | 45,867 | M | 8.6 | ✗ | ✗ | ✗ | ✓ | ✓ |
| OmniBench (Li et al., 2024d) | A+I | ✗ | ✗ | 1,142 | M | 37.8 | ✗ | ✓ | ✓ | ✓ | ✓ |
| AV-Odyssey (Gong et al., 2024) | A+I | ✗ | ✗ | 4,555 | M | 19.5 | ✗ | ✓ | ✓ | ✓ | ✓ |
| LongVALE (Geng et al., 2024) | A+V | 8,400 | 235 | ✗ | A&M | ✗ | ✗ | ✗ | ✓ | ✓ | ✓ |
| **WorldSense** | A+V | 1,662 | 141.1 | 3,172 | M | 37.2 | 986.2 | ✓ | ✓ | ✓ | ✓ |

model performance but also quantifies the models' reliance on individual modalities, highlighting the critical role of multimodal collaboration in real-world comprehension tasks.

# 4 EXPERIMENTS AND FINDINGS

## 4.1 SETTINGS

To comprehensively assess the multi-modal understanding ability, we evaluate three types of MLLMs: (1) open-source audio-visual models, such as Unified-IO-2 (Lu et al., 2024b), OneLLM (Han et al., 2024), and VideoLLaMA2 (Cheng et al., 2024); (ii) open-source MLLMs, such as Qwen2-VL (Wang et al., 2024a), LLaVA-OneVision (Li et al., 2024a), InternVL2.5 (Chen et al., 2024b), LLaVA-Video (Zhang et al., 2024d), and so on; (iii) proprietary MLLMs, such as Claude 3.5 Sonnet (Anthropic, 2024), GPT 4o (Hurst et al., 2024), Gemini 1.5 Pro (Team et al., 2024a), and Gemini 2.5 Pro (Comanici et al., 2025). For all evaluations, we strictly adhere to each model's official implementation guidelines and the recommended pre-processing procedures. Video frame extraction follows the official configurations specified by corresponding MLLMs, while proprietary models are evaluated according to their API specifications and recommended input formats. Model performance is assessed through direct comparison between model outputs and ground-truth.

## 4.2 RESULTS ON *WorldSense*

**Main Results.** We present comprehensive evaluations of *WorldSense* in Table 2. Our analysis reveals several significant insights regarding the capabilities of MLLMs in real-world understanding.

First, current open-source video models are limited in their performance as they process only visual information. This restriction highlights a significant gap in their ability to perform complex, multimodal understanding tasks, as evidenced by their maximum performance score of only 54.0%. The results underscore the inadequacies of relying solely on visual processing, emphasizing the need to integrate audio inputs for a more comprehensive understanding in practical applications.

Second and surprisingly, most of existing open-source audio-visual MLLMs perform even worse, achieving accuracy rates comparable to random guessing and notably below video-only MLLMs. This counter-intuitive finding reveals that despite having access to both modalities, these models struggle with effective audio-visual integration, suggesting that multimodal processing capability alone does not guarantee better performance without sophisticated integration mechanisms.

Table 2: **Overall performance on *WorldSense*.** We evaluate three types of MLLMs on *WorldSense*, showing the significant limitations of existing MLLMs on real-world multi-modal understanding.

| Methods | LLM Size | Tech & Science | Culture & Politics | Daily Life | Film & TV | Perfor-mance | Games | Sports | Music | Avg |
|---|---|---|---|---|---|---|---|---|---|---|
| *Open-Source Video-Audio MLLMs* | | | | | | | | | | |
| Unified-IO-2 L (Lu et al., 2024b) | 1B | 19.3 | 22.8 | 23.1 | 25.6 | 25.8 | 24.1 | 22.9 | 25.3 | 23.3 |
| Unified-IO-2 XL (Lu et al., 2024b) | 3B | 26.5 | 24.4 | 22.5 | 23.5 | 24.7 | 28.0 | 25.7 | 24.2 | 24.7 |
| Unified-IO-2 XXL (Lu et al., 2024b) | 7B | 27.1 | 31.7 | 23.9 | 23.7 | 25.5 | 23.7 | 25.7 | 27.3 | 25.9 |
| OneLLM (Han et al., 2024) | 7B | 26.7 | 25.1 | 19.0 | 22.7 | 27.0 | 23.7 | 22.4 | 19.8 | 22.8 |
| VideoLLaMA2 (Cheng et al., 2024) | 7B | 29.4 | 25.4 | 21.8 | 24.5 | 26.2 | 24.6 | 25.5 | 27.1 | 25.4 |
| VITA-1.5 (Fu et al., 2025) | 7B | 38.2 | 35.9 | 34.3 | 39.8 | 41.2 | 32.6 | 34.7 | 39.9 | 36.9 |
| Qwen2.5-Omni (Xu et al., 2025a) | 7B | 47.8 | 49.8 | 43.6 | 43.8 | 48.3 | 39.1 | 43.5 | 47.3 | 45.4 |
| video-SALMONN 2+ (Tang et al., 2025) | 7B | 57.1 | 54.4 | 48.9 | 50.9 | 49.1 | 51.1 | 44.9 | 51.0 | 50.9 |
| Qwen3-Omni (Xu et al., 2025b) | 7B | 58.7 | 60.5 | 54.5 | 53.8 | 55.4 | 46.8 | 48.8 | 52.2 | 54.0 |
| video-SALMONN 2+ (Tang et al., 2025) | 72B | 59.0 | 63.1 | 54.0 | 59.9 | 58.1 | 54.1 | 51.9 | 54.4 | 56.5 |
| *Open-Source Video MLLMs* | | | | | | | | | | |
| Video-LLaVA (Lin et al., 2023) | 7B | 23.6 | 20.8 | 19.1 | 17.3 | 23.6 | 17.2 | 20.8 | 20.1 | 20.3 |
| LLaMA3.2 (Grattafiori et al., 2024) | 7B | 27.5 | 25.7 | 28.9 | 25.9 | 27.7 | 21.1 | 29.0 | 26.8 | 27.1 |
| Qwen2-VL (Wang et al., 2024a) | 7B | 33.5 | 29.0 | 28.4 | 33.6 | 30.3 | 32.3 | 34.7 | 38.5 | 32.4 |
| mPLUG-Owl3 (Ye et al., 2024) | 7B | 37.5 | 31.4 | 31.0 | 34.1 | 33.3 | 33.2 | 32.1 | 30.5 | 32.9 |
| LLaVA-OneVision (Li et al., 2024a) | 7B | 38.9 | 38.9 | 36.3 | 37.6 | 37.8 | 37.9 | 36.3 | 39.1 | 37.7 |
| InternVL2.5 (Chen et al., 2024b) | 8B | 43.7 | 40.9 | 34.6 | 39.7 | 37.8 | 36.2 | 39.4 | 41.1 | 39.1 |
| LLaVA-Video (Zhang et al., 2024d) | 7B | 41.6 | 38.6 | 40.6 | 42.1 | 40.4 | 39.7 | 37.0 | 40.9 | 40.2 |
| *Proprietary MLLMs* | | | | | | | | | | |
| Claude 3.5 Sonnet (Anthropic, 2024) | - | 43.7 | 31.7 | 30.6 | 36.5 | 30.7 | 31.9 | 36.6 | 33.9 | 34.8 |
| GPT 4o (Hurst et al., 2024) | - | 48.0 | 44.0 | 38.3 | 43.5 | 41.9 | 41.2 | 42.6 | 42.7 | 42.6 |
| Gemini 1.5 Pro (Team et al., 2024a) | - | 53.7 | 47.2 | 50.3 | 50.4 | 52.4 | 46.8 | 40.2 | 42.0 | 48.0 |
| Gemini 2.5 Flash (Comanici et al., 2025) | - | 51.8 | 50.2 | 54.1 | 51.2 | 59.6 | 50.6 | 51.6 | 51.5 | 52.3 |
| Gemini 2.5 Pro (Comanici et al., 2025) | - | 64.9 | 66.0 | 65.8 | 68.1 | 69.7 | 65.7 | 63.5 | 61.3 | 65.1 |

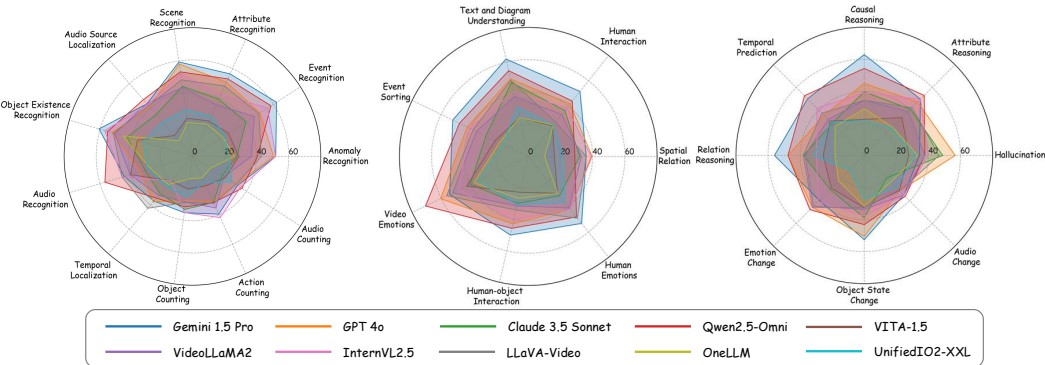

Figure 4: **Fine-grained results on task category.** We present performance across all tasks.

Third, among proprietary MLLMs, vision-only models GPT-4o and Claude 3.5 Sonnet demonstrate performance comparable to the leading open-source video MLLMs. Gemini 2.5 Pro, capable of processing both audio and visual information, achieves the highest accuracy of 65.1%. However, this performance still falls considerably short of requirements for reliable real-world applications, indicating substantial room for improvement.

These comprehensive results illuminate several critical insights: (i) the fundamental importance of audio-visual collaborative understanding in real-world scenarios; (ii) the current significant gap in models' capabilities for effective multimodal integration, and (iii) the need for more sophisticated approaches to combining and reasoning about multiple modalities. These findings point to crucial directions for future research and development in MLLMs.

**Breakdown Results.** We conduct a fine-grained analysis of model performance across different audio types and task categories, as shown in Figure 4 and 5, highlighting the limitations of MLLMs.

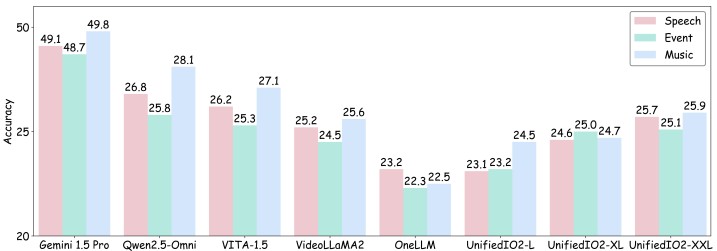
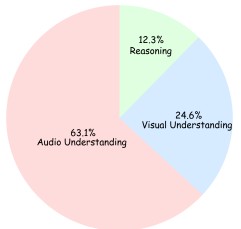

Figure 5: **Fine-grained results on audio signals.** Existing models exhibit inconsistent performance across audio types.

Figure 6: **Error distribution.** Sampled 5 error cases per task.

Table 3: **Impact of vision information.** We evaluate MLLMs' performance under different input configurations: audio-only input, audio combined with either video captions or video frames.

| Methods | Modality | Tech & Science | Culture & Politics | Daily Life | Film & TV | Perfor-mance | Games | Sports | Music | Avg |
|---|---|---|---|---|---|---|---|---|---|---|
| Unified-IO-2 L (Lu et al., 2024b) | Audio | 23.0 | 25.4 | 24.2 | 26.7 | 27.7 | 23.7 | 25.0 | 27.1 | 25.2 |
| | + Caption | 21.5 | 21.1 | 20.7 | 17.1 | 19.9 | 19.0 | 22.9 | 23.7 | $20.9_{-4.3}$ |
| | + Video | 19.3 | 22.8 | 23.1 | 25.6 | 25.8 | 24.1 | 22.9 | 25.3 | $23.3_{-1.9}$ |
| Unified-IO-2 XL (Lu et al., 2024b) | Audio | 21.7 | 22.4 | 22.4 | 22.1 | 24.7 | 25.0 | 25.9 | 24.7 | 23.4 |
| | + Caption | 19.9 | 19.8 | 20.8 | 19.2 | 20.2 | 15.9 | 21.7 | 25.5 | $20.7_{-2.7}$ |
| | + Video | 26.5 | 24.4 | 22.5 | 23.5 | 24.7 | 28.0 | 25.7 | 24.2 | $24.7_{+1.3}$ |
| Unified-IO-2 XXL (Lu et al., 2024b) | Audio | 27.5 | 28.7 | 23.9 | 23.2 | 25.8 | 21.1 | 26.2 | 30.2 | 25.9 |
| | + Caption | 24.0 | 26.7 | 23.0 | 18.9 | 18.7 | 20.7 | 25.9 | 29.4 | $23.7_{-2.2}$ |
| | + Video | 27.1 | 31.7 | 23.9 | 23.7 | 25.5 | 23.7 | 25.7 | 27.3 | $25.9_{+0.0}$ |
| OneLLM (Han et al., 2024) | Audio | 25.7 | 26.1 | 19.3 | 21.9 | 25.8 | 25.9 | 21.5 | 22.4 | 23.0 |
| | + Caption | 29.6 | 29.0 | 25.9 | 29.1 | 33.0 | 26.7 | 29.2 | 28.6 | $28.6_{+5.6}$ |
| | + Video | 26.7 | 25.1 | 19.0 | 22.7 | 27.0 | 23.7 | 22.4 | 19.8 | $22.8_{-0.2}$ |
| VideoLLaMA2 (Cheng et al., 2024) | Audio | 23.8 | 23.4 | 21.3 | 22.4 | 24.7 | 19.8 | 27.1 | 27.9 | 23.8 |
| | + Caption | 30.0 | 30.0 | 25.6 | 29.9 | 28.5 | 25.0 | 29.7 | 29.9 | $28.5_{+4.7}$ |
| | + Video | 29.4 | 25.4 | 21.8 | 24.5 | 26.2 | 24.6 | 25.5 | 27.1 | $25.4_{+1.6}$ |
| VITA-1.5 (Fu et al., 2025) | Audio | 30.2 | 35.6 | 36.3 | 30.9 | 32.2 | 32.2 | 31.4 | 33.3 | 32.9 |
| | + Caption | 39.2 | 39.8 | 37.2 | 37.5 | 37.5 | 35.2 | 34.9 | 38.4 | $37.5_{+4.6}$ |
| | + Video | 38.2 | 35.9 | 34.3 | 39.8 | 41.2 | 32.6 | 34.7 | 39.9 | $36.9_{+4.0}$ |
| Qwen2.5-Omni (Xu et al., 2025a) | Audio | 40.0 | 38.2 | 36.0 | 33.5 | 31.1 | 30.5 | 32.3 | 33.3 | 34.9 |
| | + Caption | 40.0 | 40.0 | 38.9 | 33.5 | 36.7 | 37.8 | 37.7 | 38.9 | $37.9_{+3.0}$ |
| | + Video | 47.8 | 49.8 | 43.6 | 43.8 | 48.3 | 39.1 | 43.5 | 47.3 | $45.4_{+10.5}$ |
| Gemini 1.5 Pro (Team et al., 2024a) | Audio | 40.2 | 42.9 | 35.8 | 33.3 | 33.0 | 31.0 | 33.3 | 24.7 | 34.6 |
| | + Caption | 49.5 | 52.1 | 41.8 | 42.9 | 46.4 | 41.8 | 39.6 | 36.7 | $43.6_{+9.0}$ |
| | + Video | 53.7 | 47.2 | 50.3 | 50.4 | 52.4 | 46.8 | 40.2 | 42.0 | $48.0_{+13.4}$ |

First, models consistently underperform on audio-related tasks (*e.g.*, audio recognition, audio counting) compared to other task types, demonstrating significant challenges in audio understanding. Second, spatial reasoning and counting tasks present notable difficulties for current models, a pattern consistently observed across multiple benchmarks. Third, emotion-related tasks prove particularly challenging, likely due to their requirement for integrating subtle and complex multimodal cues, including facial expressions, vocal tones, and contextual speech content. This underperformance in emotional understanding suggests a significant gap in current MLLMs' training data and capabilities, highlighting an important area for future development.

Additionally, performance varies across audio types. While Gemini 1.5 Pro performs best overall, it shows notably lower accuracy on event-related questions compared to speech or music tasks, possibly due to the complex nature of environmental sounds. Other models also exhibit inconsistent performance across audio types, underscoring a general limitation for audio understanding.

## 4.3 ROADMAP TOWARDS REAL-WORLD UNDERSTANDING

Given the substantial performance gap revealed in above evaluation, we conduct an in-depth investigation into potential approaches to enhance the MLLMs' performance.

Table 4: **Impact of audio information for Video-Audio MLLMs.** We conduct experiments across three input configurations: video-only, video with subtitles, and video with original audio.

| Methods | Speech | | | Event | | | Music | | | Overall | | |
|---|---|---|---|---|---|---|---|---|---|---|---|---|
| | Video | + Subtitle | + Audio | Video | + Subtitle | + Audio | Video | + Subtitle | + Audio | Video | + Subtitle | + Audio |
| Unified-IO-2 L (Lu et al., 2024b) | 26.8 | $13.9_{-12.9}$ | $23.1_{-3.1}$ | 26.9 | $13.5_{-13.4}$ | $23.2_{-3.7}$ | 26.3 | $15.0_{-11.3}$ | $24.5_{-1.8}$ | 26.6 | $14.8_{-11.8}$ | $23.3_{-3.3}$ |
| Unified-IO-2 XL (Lu et al., 2024b) | 25.0 | $13.0_{-12.0}$ | $24.6_{-0.4}$ | 24.8 | $12.3_{-12.5}$ | $25.0_{+0.2}$ | 26.7 | $15.9_{-10.8}$ | $24.7_{-2.0}$ | 25.3 | $14.1_{-11.2}$ | $24.7_{-0.6}$ |
| Unified-IO-2 XXL (Lu et al., 2024b) | 27.0 | $15.6_{-11.4}$ | $25.7_{-1.3}$ | 26.2 | $14.2_{-12.0}$ | $25.1_{-1.1}$ | 28.4 | $19.1_{-9.3}$ | $25.9_{-2.5}$ | 27.2 | $17.2_{-10.0}$ | $25.9_{-1.3}$ |
| OneLLM (Han et al., 2024) | 12.5 | $19.6_{+7.1}$ | $23.2_{+10.7}$ | 12.4 | $19.3_{+6.9}$ | $22.3_{+9.9}$ | 12.4 | $19.0_{+6.6}$ | $22.5_{+10.1}$ | 12.6 | $19.6_{+7.0}$ | $22.8_{+10.2}$ |
| VideoLLaMA2 (Cheng et al., 2024) | 17.1 | $25.5_{+8.4}$ | $25.2_{+8.1}$ | 16.1 | $24.9_{+8.8}$ | $24.5_{+8.4}$ | 17.7 | $27.0_{+9.3}$ | $25.6_{+7.9}$ | 17.4 | $26.1_{+8.7}$ | $25.4_{+8.0}$ |
| VITA-1.5 (Fu et al., 2025) | 37.6 | $39.1_{+1.5}$ | $36.2_{-1.4}$ | 36.4 | $38.2_{+1.8}$ | $35.3_{-1.1}$ | 38.7 | $40.0_{+1.3}$ | $37.1_{-1.6}$ | 37.7 | $39.3_{+1.6}$ | $36.5_{-1.2}$ |
| Qwen2.5-Omni (Xu et al., 2025a) | 38.7 | $38.7_{+0.0}$ | $44.8_{+6.1}$ | 37.6 | $37.7_{+0.1}$ | $43.8_{+6.2}$ | 40.7 | $40.3_{-0.4}$ | $46.1_{+5.4}$ | 39.2 | $39.2_{+0.0}$ | $45.2_{+6.0}$ |
| Gemini 1.5 Pro (Team et al., 2024a) | 34.3 | $39.6_{+5.3}$ | $49.2_{+14.9}$ | 33.0 | $38.9_{+5.9}$ | $48.7_{+15.7}$ | 35.4 | $39.2_{+3.8}$ | $49.8_{+14.4}$ | 34.4 | $39.3_{+4.9}$ | $48.0_{+13.6}$ |

Table 5: **Impact of audio information for Video MLLMs.** We provide video-only MLLMs with the subtitles and compare the performance with models with only video input.

| Methods | Speech | | Event | | Music | | Overall | |
|---|---|---|---|---|---|---|---|---|
| | Video | + Subtitle | Video | + Subtitle | Video | + Subtitle | Video | + Subtitle |
| Video-LLaVA (Lin et al., 2023) | 20.3 | $15.4_{-4.9}$ | 19.8 | $14.4_{-5.4}$ | 19.5 | $16.4_{-3.1}$ | 20.3 | $16.0_{-4.3}$ |
| LLaMA3.2 (Grattafiori et al., 2024) | 27.1 | $29.3_{+2.2}$ | 27.6 | $29.6_{+2.0}$ | 25.9 | $28.1_{+2.2}$ | 27.1 | $28.8_{+1.7}$ |
| Qwen2-VL (Wang et al., 2024a) | 31.8 | $41.1_{+9.3}$ | 30.9 | $39.4_{+8.5}$ | 34.2 | $41.8_{+7.6}$ | 32.4 | $41.2_{+8.8}$ |
| mPLUG-Owl3 (Ye et al., 2024) | 33.0 | $39.2_{+6.2}$ | 32.3 | $38.3_{+6.0}$ | 34.6 | $39.2_{+4.6}$ | 32.9 | $38.7_{+5.8}$ |
| LLaVA-OneVision (Li et al., 2024a) | 37.7 | $44.0_{+6.3}$ | 36.3 | $42.7_{+6.4}$ | 39.7 | $45.7_{+6.0}$ | 37.7 | $43.9_{+6.2}$ |
| InternVL2.5 (Chen et al., 2024b) | 39.0 | $48.3_{+9.3}$ | 38.6 | $47.9_{+9.3}$ | 39.2 | $47.1_{+7.9}$ | 39.1 | $47.8_{+8.7}$ |
| LLaVA-Video (Zhang et al., 2024d) | 40.5 | $45.9_{+5.4}$ | 38.9 | $44.6_{+5.7}$ | 42.3 | $47.7_{+5.4}$ | 40.2 | $45.6_{+5.4}$ |
| GPT 4o (Hurst et al., 2024) | 42.8 | $51.1_{+8.3}$ | 40.9 | $50.2_{+9.3}$ | 43.6 | $49.9_{+6.3}$ | 42.6 | $50.1_{+7.5}$ |

**Vision Information.** We investigate the impact of visual information through different input configurations: audio-only, audio with video captions, and audio with video frames. As shown in Table 3, visual information generally improves performance, with Gemini 1.5 Pro's accuracy increasing from $34.6\%$ (audio-only) to $48.0\%$ (+video). However, impact varies across models, with UnifiedIO2 showing inconsistent gains and even degradation with captions.

These findings suggest two important insights: (1) visual information is crucial for enhancing multimodal understanding when properly integrated, and (2) current models' ability to effectively utilize visual information remains limited.

**Audio Information.** We examine the impact of audio information through three configurations: video-only, video with subtitles, and video with original audio.

The results in Table 4 reveal intriguing patterns in how different forms of audio information influence model performance. For Gemini 1.5 Pro, accuracy increases from $34.4\%$ (video-only) to $39.3\%$ with subtitles, and further to $48.0\%$ with original audio. Other models, such as OneLLM and Qwen2.5-Omni, show similar improvements. These results demonstrate that both subtitles and acoustic features (including tone, emotion, and environmental sounds) contribute valuable information, beyond what subtitles alone can capture, emphasizing the importance of complete acoustic cues in omni-modal real-world understanding.

Interestingly, UnifiedIO2 demonstrates performance degradation when integrating either subtitles or audio, with subtitles causing a notable accuracy decline, suggesting difficulties in multimodal processing. Conversely, Video-LLaMA2 improves with both modalities but performs better with subtitles than original audio, indicating stronger reliance on textual rather than acoustic information.

We further evaluate video-only MLLMs by providing transcribed subtitles, as shown in Table 5. Nearly all models show significant improvements with subtitle integration, reinforcing the importance of audio information. However, the performance gain is less pronounced in music-related questions, as subtitles cannot effectively capture inherent acoustic features such as melody, rhythm, and harmony.

These evaluations highlight several critical findings: (i) original audio contains rich information beyond what subtitles can capture, particularly for music; (ii) current models show significant limitations in multimodal processing. These insights suggest important directions for improving MLLMs' ability to integrate acoustic and textual information for comprehensive scene understanding.

**Failure Analysis and Future Improvement.** We perform error analysis on 130 samples of Gemini 1.5 Pro (5 random samples per task) through manual review, identifying three main error types: Au-

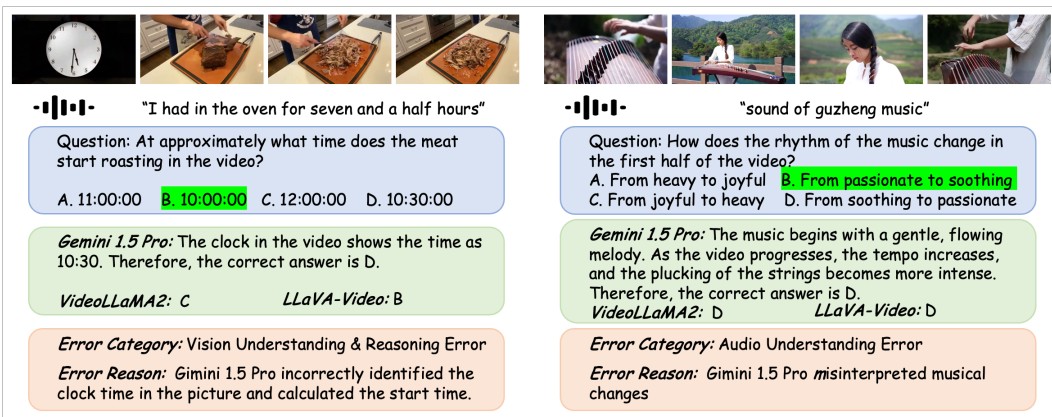

Figure 7: **Failure Case.** We present two error examples.

dio Understanding Errors (misinterpreting audio information), Visual Understanding Errors (missing visual details), and Reasoning Errors (faulty logical steps). As shown in Figure 6, most errors stem from audio understanding deficiencies and reasoning failures. The reason for poor accuracy and limitation of existing models can be summarized as follows: (i) **Inadequate Audio Understanding.** Existing models fail to understand audio information correctly and show significantly weaker audio processing than visual understanding. (ii) **Limited Cross-Modal Integration.** Models often process modalities independently rather than performing true multimodal integration and suffer from insufficient omni-modal information integration. (iii) **Insufficient Complex Reasoning Ability.** Despite correct perception, MLLMs still conduct error reasoning, leading to incorrect conclusions.

Figure 7 showcases two failure cases of Gemini 1.5 Pro, VideoLLaMA2, and LLaVA-Video. The left case involves a vision understanding and reasoning error, where Gemini 1.5 Pro incorrectly identified the clock time as 10:30 instead of the actual 10:00 displayed, leading to the incorrect answer. This reflects deficiencies in basic visual perception and properly correlating visual information with the question. The right case demonstrates an audio understanding error, where models misjudged the rhythm pattern change in guzheng music, interpreting it as changing from soothing to intense (option D) rather than the correct passionate to soothing (option B). This case indicates that tasks involving interpretation of emotion and rhythm patterns remain challenging for existing MLLMs.

We also raise several key strategies to enhance models' understanding of omni-modality information: (i) **Coupled Multimodal Training Data.** Using naturally coupled, interleaved multimodal data, for example, audio, visual, language content, would enhance models' capability to leverage cross-modal dependencies. (ii) **Architectural Improvements.** Enhanced attention mechanisms facilitating deep multimodal integration could emphasize early fusion between modalities, rather than processing them as separate streams for late fusion. (iii) **Advanced Modal Alignment Techniques.** Progressive alignment strategies that gradually enhance the model's ability to align information across modalities could lead to more effective utilization of multimodal inputs. (iiii) **Reasoning strengthening.** Incorporating diverse reasoning-focused data can strengthen logical inference capabilities, enabling more coherent and accurate conclusions.

## 5 CONCLUSION

In this paper, we propose *WorldSense*, the ***first*** benchmark designed to evaluate MLLMs' omni-modal understanding in real-world scenarios. Distinguished by its emphasis on joint omnimodal comprehension across diverse real-world contexts, *WorldSense* encompasses rich video categories and carefully curated question-answer pairs that necessitate the integration of visual and acoustic information. Through extensive experiments, we expose significant limitations in current MLLMs' ability to process and coherently integrate omnimodal information. Our analysis demonstrates the importance of omnimodal collaboration in real-world understanding. We hope that *WorldSense* can serve as a foundational benchmark for advancing human-like omnimodal understanding capabilities.

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

# A APPENDIX

## A.1 THE USE OF LARGE LANGUAGE MODELS (LLMS)

Authors use large language models (LLMs) solely as a writing assistant for text refinement and language polishing.

## A.2 QUALITY CONTROL

**Experienced annotators.** Our annotation team consists of 80 professional annotators with extensive QA annotation experience. These annotators are proficient in English, and have participated in sereval QA data annotation projects.

**Sufficient annotation training.** We conducted a one-week training program with 200 videos (excluded from final benchmark) until annotators achieved high proficiency (only $10\%$ requiring modifications).

**Annotation instruction.** Each annotator received a comprehensive instruction with task explanations, question formulation guidelines, QA creation instructions, annotated examples, and cross-modal inference requirements.

Our review process identified and revised similar QA pairs. Through **professional annotators**, **thorough training**, **detailed guidelines**, and r**igorous quality control**, we ensure high-quality annotations.

## A.3 IMPLEMENT DETAILS

For open-source MLLMs, we strictly follow their official implementations and recommended pre-processing pipelines to ensure fair comparison. For GPT 4o and Claude 3.5 Sonnet, we sample 16 frames uniformly from each video, while for Gemini 1.5 Pro, we utilize the official API for raw video file uploads. We conduct all the experiments on a NVIDIA A100 GPU.

## A.4 EVALUATION PROMPT

Following previous works (Fu et al., 2024b; Li et al., 2024c), we adopt the format of "whole video frames + whole subtitles/audios (optional) + question with prompt" as prompt. We show the evaluation prompt across three input configurations: video-only input, video with subtitles, and video with audio content as following.

**Evaluation Prompt**

```
Carefully watch this video and pay attention to every detail.
Based on your observations, select the best option that accurately
addresses the question.

These are the frames of a video.  Select the best answer to the
following multiple-choice question based on the video.  Respond
with only the letter (A, B, C, or D) of the correct option.
```

**Question:** {}
{Option1}
{Option2}
{Option3}
{Option4}
**Answer:**

**Evaluation Prompt with Subtitles**

```
Carefully watch this video and pay attention to every detail.
Based on your observations, select the best option that accurately
addresses the question.

These are the frames of a video.  This video's subtitles are listed
below:

{subtitles}

Select the best answer to the following multiple-choice question
based on the video.  Respond with only the letter (A, B, C, or D)
of the correct option.
```

**Question:** {}
{Option1}
{Option2}
{Option3}
{Option4}
**Answer:**

**Evaluation Prompt with Audios**

```
Carefully watch this video and pay attention to every detail.
Based on your observations, select the best option that accurately
addresses the question.

These are the frames of a video and the corresponding audio.
Select the best answer to the following multiple-choice question
based on the video.  Respond with only the letter (A, B, C, or D)
of the correct option.
```

**Question:** {}
{Option1}
{Option2}
{Option3}
{Option4}
**Answer:**

## A.5 LIMITATION

While our WorldSense represents a significant advancement in evaluating multimodal understanding capabilities of MLLMs, the multiple-choice format inevitably constrains the assessment of models' generative capabilities. Real-world understanding often requires open-ended responses, explanations, and adaptability beyond selecting from predefined options. Our WorldSense may not adequately evaluate how models perform on tasks requiring nuanced reasoning or creative problem-solving. We will add open-ended questions and expand the evaluation paradigm to better assess real-world multimodal understanding.

## B BROADER IMPACTS & ETHICS STATEMENT

Our work on WorldSense has several potential positive impacts on society and AI development, while also presenting certain risks that warrant careful consideration. WorldSense contributes to advancing MLLMs' ability to understand and interact with the real world through multiple modalities. This progress could benefit various applications, including assistive technologies, educational tools, human-AI interaction systems, safety systems, and so on. We also acknowledge potential risks and challenges. The development of more capable AI systems might raise privacy concerns. Advanced multimodal understanding capabilities could potentially be misused for surveillance or monitoring purposes. We believe that open discussion of these impacts is crucial for the responsible development of multi-modal large language models.

Our research on WorldSense adheres to strict ethical principles and guidelines. We acknowledge several important ethical considerations: (1) **Data Collection and Privacy.** All video content in WorldSense has been collected from publicly available sources with appropriate licensing agreements. We have con ducted thorough reviews and implemented comprehensive data processing procedures to ensure privacy protection, including the removal of any personally identifiable information. (2) **Potential Biases.** While acknowledging that inherent biases may exist in any dataset, we have undertaken systematic efforts to ensure diverse representation across our video content and question-answer pairs, encompassing various domains, cultures, and contexts. Nevertheless, we recognize that completely eliminating bias remains a significant challenge, and users should carefully consider these potential limitations when utilizing our dataset. (3) **Intended Use.** WorldSense is specifically designed to advance research in omnimodal real-world understanding. While we actively encourage the use of this benchmark for academic and research purposes, we strongly caution against any applications that could potentially result in harmful or discriminatory outcomes. Users are expected to adhere to ethical guidelines and responsible practices.

## B.1 LICENSE

The WorldSense dataset is released under the CC BY-NC-SA 4.0 License. Authors bear all responsibility in case of violation of rights and confirmation of the data license.

## B.2 DATASHEETS

## B.3 MOTIVATION

- **For what purpose was the dataset created?**

  To evaluate MLLMs' capabilities in real-world omnimodal understanding.

- **Who created the dataset (e.g., which team, research group) and on behalf of which entity (e.g., company, institution, organization)?**

  The authors of this paper.

- **Who funded the creation of the dataset?**

  Xiaohongshu Inc.

- **Any other comments?**

  No

## B.4 COMPOSITION

- **What do the instances that comprise the dataset represent (e.g., documents, photos, people, countries)?**

  Videos along with captions and question/answer pairs.

- **How many instances are there in total (of each type, if appropriate)?**

  WorldSense contains 3,172 question-answer pairs and contains 1,662 videos in total.

- **Does the dataset contain all possible instances or is it a sample (not necessarily random) of instances from a larger set?**

  Videos of WorldSense are sampled from FineVideo and Music AVQA. All QA pairs are re-annotated manually.

- **What data does each instance consist of?**

  Each instance contains one video with its corresponding audio, a question about the video content and the corresponding answer, the category of the video, the fine-grained video understanding capability examined by the question, and the class of audio content. Each instance also contain the auto-generated subtitles sourced from YouTube.

- **Is there a label or target associated with each instance?**

  Yes. We provide the ground-truth answer for each question.

- **Is any information missing from individual instances?**

  N/A.

- **Are relationships between individual instances made explicit (e.g., users' movie ratings, social network links)?**

  N/A.

- **Are there recommended data splits (e.g., training, development/validation, testing)?**

  No, WorldSense is designed for evaluation only.

- **Are there any errors, sources of noise, or redundancies in the dataset?**

  No.

- **Is the dataset self-contained, or does it link to or otherwise rely on external resources (e.g., websites, tweets, other datasets)?**

  WorldSense is self-contained.

- **Does the dataset contain data that might be considered confidential (e.g., data that is protected by legal privilege or by doctor – patient confidentiality, data that includes the content of individuals' non-public communications)?**

  N/A.

- **Does the dataset contain data that, if viewed directly, might be offensive, insulting, threatening, or might otherwise cause anxiety?**

  N/A.

## B.5 COLLECTION PROCESS

- **How was the data associated with each instance acquired?**

  See main paper for details.

- **What mechanisms or procedures were used to collect the data (e.g., hardware apparatuses or sensors, manual human curation, software programs, software APIs)?**

  Humans are required to propose a question and corresponding answer based on the video. MLLMs, such as Qwen2-VL, Video-LLaMA2 and OneLLM are utilized to perform quality control.

- **If the dataset is a sample from a larger set, what was the sampling strategy (e.g., deterministic, probabilistic with specific sampling probabilities)?**

  Yes, we sample the videos from FineVideo and Music-AVQA. See main paper for details.

- **Who was involved in the data collection process (e.g., students, crowdworkers, contractors) and how were they compensated (e.g., how much were crowdworkers paid)?**
  The authors and contractors are involved in the data collection process and are paid a fair wage.

- **Over what timeframe was the data collected?**
  The dataset is collected in 2024.

- **Were any ethical review processes conducted (e.g., by an institutional review board)?**
  All videos in our benchmark are human-selected based on appropriate value propositions and undergo a second manual quality check to ensure there are no ethical violations.

- **Did you collect the data from the individuals in question directly, or obtain it via third parties or other sources (e.g., websites)?**
  We obtained video data from FineVideo and Music-AVQA.

- **Were the individuals in question notified about the data collection?**
  We didn't collect the data from the individuals. The data was collected from public web sources instead.

- **Did the individuals in question consent to the collection and use of their data?**
  N/A.

- **If consent was obtained, were the consenting individuals provided with a mechanism to revoke their consent in the future or for certain uses?**
  N/A.

- **Has an analysis of the potential impact of the dataset and its use on data subjects (e.g., a data protection impact analysis) been conducted?**
  N/A.

- **Any other comments?**
  No.

## B.6    PREPROCESSING/CLEANING/LABELING

- **Was any preprocessing/cleaning/labeling of the data done (e.g., discretization or bucketing, tokenization, part-of-speech tagging, SIFT feature extraction, removal of instances, processing of missing values)?**
  We firstly select videos based on pre-designed categories, and then clip the video based on visual-audio correlation and dynamic scores.

- **Was the "raw" data saved in addition to the preprocessed/cleaned/labeled data (e.g., to support unanticipated future uses)?**
  N/A.

- **Is the software that was used to preprocess/clean/label the data available?**
  We use the open-source models.

- **Any other comments?**
  No.

## B.7    USES

- **Has the dataset been used for any tasks already?**
  Yes. We have used the dataset to evaluate video question answering in real-world.

- **Is there a repository that links to any or all papers or systems that use the dataset?**
  No.

- **What (other) tasks could the dataset be used for?**
  It also can be used to evaluate the video understanding capability of VLMs.

- **Is there anything about the composition of the dataset or the way it was collected and preprocessed/cleaned/labeled that might impact future uses?**
  No.

- **Are there tasks for which the dataset should not be used?**
  N/A.
- **Any other comments?**
  No.

## B.8 DISTRIBUTION

- **Will the dataset be distributed to third parties outside of the entity (e.g., company, institution, organization) on behalf of which the dataset was created?**
  Yes, the dataset will be made publicly available.
- **How will the dataset will be distributed (e.g., tarball on website, API, GitHub)?**
  We host it on the webpage, GitHub, and Huggingface.
- **When will the dataset be distributed?**
  It's availale and open to the public now.
- **Will the dataset be distributed under a copyright or other intellectual property (IP) license, and/or under applicable terms of use (ToU)?**
  We release our benchmark under CC BY-NC 4.0 license.
- **Have any third parties imposed IP-based or other restrictions on the data associated with the instances?**
  No.
- **Do any export controls or other regulatory restrictions apply to the dataset or to individual instances?**
  No.
- **Any other comments?**
  No.

## B.9 MAINTENANCE

- **Who will be supporting/hosting/maintaining the dataset?**
  The authors will be supporting/hosting/maintaining the dataset.
- **How can the owner/curator/manager of the dataset be contacted (e.g., email address)?**
  No.
- **Is there an erratum?**
  Currently, we do not have an erratum. We will update if we find errors.
- **Will the dataset be updated (e.g., to correct labeling errors, add new instances, delete instances)?**
  Yes. We will make announcements on GitHub if there is any update.
- **If the dataset relates to people, are there applicable limits on the retention of the data associated with the instances (e.g., were the individuals in question told that their data would be retained for a fixed period of time and then deleted)?**
  N/A.
- **Will older versions of the dataset continue to be supported/hosted/maintained?**
  Yes.
- **If others want to extend/augment/build on/contribute to the dataset, is there a mechanism for them to do so?**
  Yes. Contributors can post issues or submit pull requests on GitHub. We will review and verify contributions, and update the dataset if the contribution is useful.
- **Any other comments?**
  No.

