# OpenReview forum: "WorldSense: Evaluating Real-world Omnimodal Understanding for Multimodal LLMs"
_ICLR.cc/2026/Conference — ICLR 2026 Poster_

### Official Review · Reviewer_E5me · 2025-10-27

**Soundness:** 3
**Presentation:** 3
**Contribution:** 3
**Rating:** 4
**Confidence:** 5

**Summary:**

This paper introduces WorldSense, a new benchmark for evaluating the omnimodal understanding capabilities of multimodal LLMs. The benchmark simultaneously incorporates visual, audio, and text inputs to assess a model's ability to comprehend and reason about real-world scenarios. The paper also presents a detailed evaluation of existing multimodal LLMs on the WorldSense benchmark.

**Strengths:**

- The WorldSense Benchmark seems well annotated and comprehensively considers visual and auditory information, which is meaningful for evaluating a model's ability to understand video.

- The experiments are sufficient and convincing.

**Weaknesses:**

- In terms of task design, although WorldSense includes a list of tasks, the paper does not clearly show the special design targeted at the integration of audio and visual information. To me, it seems many tasks either assess audio understanding independently or visual understanding independently, rather than evaluating the understanding of audio and video as a whole.

-  There are already some benchmarks for evaluating audio-visual video understanding, such as AVUT [1] and DailyOmni [2]. The paper did not discuss or compare with them. What are the main differences between WorldSense and these benchmarks?

[1] Yang et al, Audio-centric Video Understanding Benchmark without Text Shortcut, arXiv preprint arXiv:2503.19951

[2] Zhou et al, Daily-Omni: Towards Audio-Visual Reasoning with Temporal Alignment across Modalities, arXiv preprint arXiv:2505.17862

**Questions:**

- About the evaluation settings.  In Line 316, the paper mentions that the models are tested "following the recommended pre-processing procedures". What are the specific test settings for each model (such as the frame rate for video frame extraction, the maximum number of extracted frames, etc.)? Could the different pre-processing settings between models lead to incomparable results?

- The audio-visual LLMs evaluated in the paper seem a bit weak on WorldSense. Could the authors provide an analysis of recent powerful omni models like Qwen3-Omni and Video-SALMONN 2? This would establish a more robust and current baseline, and also help to verify if the significant challenges highlighted by the benchmark are persistent even for the latest generation of models.

---

> ### Author Response · Authors · 2025-11-20
> **Response to Reviewer E5me - 1 / 2**
>
> Thank you for recognizing the value of our work and the sufficient experiments. We hope our response can address your concerns. We would appreciate it very much if you could reconsider our work and support us.
>
>
>
> > **Q1: In terms of task design, although WorldSense includes a list of tasks, the paper does not clearly show the special design targeted at the integration of audio and visual information. To me, it seems many tasks either assess audio understanding independently or visual understanding independently, rather than evaluating the understanding of audio and video as a whole.**
>
> The core objective of Worldsense is to evaluate multimodal models’ understanding capabilities in omnimodal scenarios (video and audio synergy). Therefore, we have implemented a series of measures to ensure our QA tasks assess models’ ability to understand video-audio synergy rather than a single modality. Additionally, all other reviewers have also recognized that WorldSense requires the integration of audio and video for correct answers, rather than relying on a single modality alone.
>
>
> 1. **Examples.** We present two QA examples from Worldsense in ***Figure 1***. These examples clearly require full integration of video and audio for correct answers. Missing either modality makes accurate responses impossible.
>
> 2. **Annotation Pipeline.**
> Through our carefully designed annotation pipeline (***Section 3.3***), we ensure all QA tasks demand both audio and video information.
>
> - We require professional annotators to generate QA pairs for each video that can only be answered via the integration of multiple modalities.
>
> - We adopt strict quality control: multimodal large models are used for automatic verification to exclude questions answerable solely with visual information.
>
> - We also remove overly simple questions to maintain task challenge.
>
> 3. **Experimental Validation of Multimodal Requirement.**
>
> Our experiments further confirm that QAs in WorldSense rely on video-audio synergy rather than a single modality. For instance, in ***Tables 3, 4, and 5***, models incorporating additional modalities achieve significant performance improvements. This fully demonstrates that these questions require the collaboration of multimodal information, rather than relying on a single modality alone.
>
> In summary, we design the **annotation pipeline** to ensure that questions require multimodal integration for correct answers. We provide **question examples** to illustrate this requirement. Furthermore, **extensive experiments** demonstrate that QAs in WorldSense demand the collaboration of multiple modalities rather than relying solely on a single modality.
>
>
>
>
> > **Q2: There are already some benchmarks for evaluating audio-visual video understanding, such as AVUT [1] and DailyOmni [2]. The paper did not discuss or compare with them. What are the main differences between WorldSense and these benchmarks?**
>
> WorldSense differs significantly from previous works.
>
> 1. First, Worldsense is the ***first*** benchmark proposed to evaluate multimodal models’ omnimodal understanding capabilities.
>
> 2. **DailyOmni.** Worldsense has a larger scale compared to Daily-Omni. Daily-Omni contains only 1,197 QA pairs, while Worldsense has 3,172 (nearly **three times** that of Daily-Omni). Worldsense covers broader video categories and task types.
>
> 3. **AVUT.** AVUT's task categories are mainly audio-centric, while Worldsense not only contains most of AVUT’s categories but also many more, resulting in a substantially broader scope.
>
>
> In summary, Worldsense has **a larger dataset size** and covers **broader video and question categories** than previous works.
>
>
>
> > **Q3: About the evaluation settings. In Line 316, the paper mentions that the models are tested "following the recommended pre-processing procedures". What are the specific test settings for each model (such as the frame rate for video frame extraction, the maximum number of extracted frames, etc.)? Could the different pre-processing settings between models lead to incomparable results?**
>
> We have detailed the experimental settings in ***Appendix A.3***. For GPT-4o and Claude 3.5 Sonnet, we uniformly sample 16 frames, which is consistent with many previous benchmarks such as VideoMME. For Gemini, we uploaded raw video files via its official API. For open-source models, we adopted their official configurations to ensure fair comparison.

---

> ### Author Response · Authors · 2025-11-20
> **Response to Reviewer E5me - 2 / 2**
>
> > **Q4: The audio-visual LLMs evaluated in the paper seem a bit weak on WorldSense. Could the authors provide an analysis of recent powerful omni models like Qwen3-Omni and Video-SALMONN 2? This would establish a more robust and current baseline, and also help to verify if the significant challenges highlighted by the benchmark are persistent even for the latest generation of models.**
>
> We sincerely apologize that we failed to evaluate the latest models in our work. As shown in the following table, we report the performance of the latest models on WorldSense. Despite outperforming earlier models, their performance remains poor, which underscores both Worldsense’s challenging characteristics and the limitations of existing open-source models. We will add the performance comparison of the latest models in the revised version.
>
>
> | Model | Average |
> | --- | --- |
> | Qwen3 Omni | 54.0 |
> | OmniVinci | 46.2 |
> | video-SALMONN 2+ 7B | 50.9 |
> | video-SALMONN 2+ 72B | 56.5 |
> | Qwen2.5 Omni | 45.4 |

---

> > ### Comment · Reviewer_E5me · 2025-11-26
> >
> > Thanks for the rebuttal. My concerns have been addressed. Please include the new results in the paper. I will raise my score.

---

> > > ### Author Response · Authors · 2025-11-26
> > >
> > > Thanks for your insightful suggestions. We have added the performance of latest models in Table 2, updated the PDF, and highlighted the caption in blue font. We also appreciate your support for our work. We thanks for your raising the scores. Your constructive comments are of great significance to our paper.

---

### Official Review · Reviewer_zBjg · 2025-11-01

**Soundness:** 3
**Presentation:** 2
**Contribution:** 3
**Rating:** 6
**Confidence:** 4

**Summary:**

This paper proposes WorldSense, a benchmark designed for omni-modal video understanding. Specifically, to answer questions in WorldSense, an MLLM’s response must rely on both video and audio information. Experiments on the proposed WorldSense benchmark reveal the limitations of current MLLMs in omni-modal reasoning.

**Strengths:**

1. Requiring both video and audio modalities for accurate responses to each question in WorldSense facilitates a more comprehensive evaluation of current MLLMs in omni-modal reasoning.
2. Experimentally, the performance drop of current video-audio MLLMs indicates that the fusion between modalities is ineffective or even detrimental.

**Weaknesses:**

1. While WorldSense emphasizes real-world omni-modal perception, understanding, and reasoning, the benchmark primarily consists of QA pairs. I believe that interactive question answering would be more practical for real-world scenarios. Moreover, isn’t the term "omni-modal" somewhat overstated, given that the benchmark only includes video, audio, and text modalities?
2. Lack of analysis on why the fusion of open-source audio and video models failed. While I wouldn’t tend to reject the paper for this reason, providing such an analysis would offer the community deeper insights than merely presenting the conclusion.

**Questions:**

1. #82–83: There’s a typo ,“THe”.
2. #129–130: Pay attention to the spacing between the image title and the main text. It currently looks a bit confusing.

---

> ### Author Response · Authors · 2025-11-20
> **Response to Reviewer zBjg**
>
> We sincerely appreciate the reviewer's positive comments on our motivation and experiments, along with your thoughtful feedback and suggestions. We hope our response can address your concerns.
>
>
> > **Q1: While WorldSense emphasizes real-world omni-modal perception, understanding, and reasoning, the benchmark primarily consists of QA pairs. I believe that interactive question answering would be more practical for real-world scenarios. Moreover, isn’t the term "omni-modal" somewhat overstated, given that the benchmark only includes video, audio, and text modalities?**
>
> Thanks for your insightful comments.
>
> 1. **Following Previous Works.** Considering that Worldsense is the ***first*** benchmark designed to evaluate omnimodal understanding, we adopt the QA format by following previous works such as VideoMME. This format is more straightforward to evaluate and implement.
>
> 2. **Poor Interaction Ability.** Additionally, given that existing models perform poorly in streaming-based responses, we do not opt for interactive question answering.
>
> We will further expand our question types and add interactive QA in future work to make Worldsense more applicable to practical use cases. Besides, we will follow your advice and replace "omnimodal" into "multimodal".
>
>
> > **Q2: Lack of analysis on why the fusion of open-source audio and video models failed. While I wouldn’t tend to reject the paper for this reason, providing such an analysis would offer the community deeper insights than merely presenting the conclusion.**
>
> Thanks for your suggestion. We have provided the error distribution in ***Figure6*** and some failure cases in ***Appendix Figure 7***. Additionally, we conduct an in-depth analysis of the reasons for fusion failures in ***Section 4.3***. The causes of fusion failures in existing models can be categorized into three types: **Inadequate Audio Understanding**, **Limited Cross-Modal Integration**, and **Insufficient Complex Reasoning Ability**. We will further emphasize the reasons for these errors in the revised manuscript.
>
>
> > **Q3: Typo**
>
> We will carefully check our manuscript, correct typos, and reformat it.

---

> > ### Comment · Reviewer_zBjg · 2025-11-26
> >
> > Thanks for the rebuttal. Please make sure to clarify Q2 and the other questions in the final version. I will keep my rating and tend to accept this paper.

---

> > > ### Author Response · Authors · 2025-11-27
> > >
> > > We are grateful for the your constructive feedback, which has helped us significantly improve the manuscript. Following your suggestions, we have added a detailed discussion in ***Lines 462-471 of the revised paper***. For the your convenience, these additions and all other substantial changes are marked with blue font.

---

### Official Review · Reviewer_Fj9V · 2025-11-01

**Soundness:** 4
**Presentation:** 3
**Contribution:** 4
**Rating:** 10
**Confidence:** 5

**Summary:**

This paper propose WorldSense, an omnimodal benchmark including video, audio, and text. WorldSense requires joint reasoning over synchronized visual and audio inputs to evaluate the ability of existing MLLMs. It contains 3,172 MC QA pairs, spanning various domains and subcategories. The paper evaluates open-source and proprietary MLLMs on WorldSense, showing that current MLLMs are having trouble in integrating audio and visual data effectively in realistic setting.

**Strengths:**

1. Solid motivation. The paper focus on omnimodal reasoning, requiring the models to utilize visual and audio inputs together to answer the question. This designed correlations make it distinct from most existing MLLM benchmarks.
2. High-quality human-reviewed annotations. Unlike recent benchmarks, WorldSense is reviewed and revised by human expert, instead of relying solely on LLMs. This is a guarantee for the quality of the benchmark. It is very rare these days.
3. Comprehensive experiments. The paper conducted comprehensive experiments on existing models, including open-source ones and proprietary ones, providing a benchmarking foundation for future research.

**Weaknesses:**

1. Video caption is not a good representative for "text" modality. This makes the "omni" a little overclaim. Given real-world constraints, audio–video may already suffice for evaluating omnimodality, as seen in emerging “world models.”
2. The question types are restricted in multiple-choice QA. This is common in existing benchmarks, but given the ability of MLLMs, free-form answers can yield deeper insights and align with user-end usage.
3. The benchmark is currently focusing on perception and recognition. Given the lengths of the dataset (141.1s) and the requirement for multiple modalities, it may be possible to curate a subset for higher-level reasoning tasks.

**Questions:**

1. How does WorldSense handle temporal reasoning—do questions depend on sequential context or only short clips? An ablation study on image / a few frames would elaborate this.
2. This is beyond the scope of this paper. But is it possible to extend WorldSense to an open-ended or generative benchmark? This would make it more aligned with user-end usage.

---

> ### Author Response · Authors · 2025-11-20
> **Response to Reviewer Fj9V**
>
> Thanks for your insightful feedback and support. We sincerely appreciate your valuable comments and your recognition of the novelty of our benchmark and the soundness of the experiments.
>
>
> > **Q1: Video caption is not a good representative for "text" modality. This makes the "omni" a little overclaim. Given real-world constraints, audio–video may already suffice for evaluating omnimodality, as seen in emerging “world models.”**
>
> It seems there is a certain misunderstanding regarding the evaluation method of WorldSense. The core evaluation approach of Worldsense is precisely the ***audio-video*** method you mentioned. For instance, the Open-Source Video-Audio MLLMs in ***Table 2*** are evaluated through this ***audio-video*** approach without providing video captions. In ***Table 3***, we only use video captions to verify the importance of visual information for omnimodality understanding. We will further clarify our evaluation method in the revised manuscript.
>
> > **Q2: The question types are restricted in multiple-choice QA. This is common in existing benchmarks, but given the ability of MLLMs, free-form answers can yield deeper insights and align with user-end usage.**
>
> Thanks for your suggestion. We opt for multiple-choice format primarily to follow prior works, and this format also offers the advantage of easy evaluation.
>
> 1. **Following Previous Works.** When constructing our WorldSense, we follow widely used benchmarks such as Video-MME, which only use multiple-choice questions for evaluation, so we also use multiple-choice questions for assessment.
>
> 2. **Easy to Assess.** We choose to evaluate on multiple-choice QA, as they can be easily assessed compared to open-ended questions, reducing the need for manual evaluation and association. Besides, the open-ended questions require an LLM-judge for evaluation, which introduces the reliability concerns and potential biases.
>
> 3. **Easy convert to open-end questions.** WorldSense could be readily adapted to open-ended formats by presenting only the questions without options and using the correct choice as the reference answer.
>
> Thank you for your suggestion. We will annotate open-ended questions in the future to achieve a more comprehensive evaluation.
>
> > **Q3: The benchmark is currently focusing on perception and recognition. Given the lengths of the dataset (141.1s) and the requirement for multiple modalities, it may be possible to curate a subset for higher-level reasoning tasks.**
>
> Thanks for your suggestion. In fact, Worldsense already includes reasoning-related questions. The number of reasoning questions is presented in ***Figure 2***. Additionally, ***Figure 4*** shows the accuracy for each task type. Existing models perform worst on reasoning tasks, indicating that these models have significant limitation in reasoning.
>
>
> > **Q4: How does WorldSense handle temporal reasoning—do questions depend on sequential context or only short clips? An ablation study on image / a few frames would elaborate this.**
>
> We think that questions that can only be answered using the full video assess a model’s holistic understanding, whereas questions that can only be answered from a short clip assess its fine-grained perception. In fact, our questions may rely on an entire video or a specific short clip. We have not imposed restrictions on this aspect. We intentionally avoid explicit limitations during the annotation process to ensure the questions are more diverse and challenging, enabling a comprehensive evaluation of model’s capabilities.
>
>
> > **Q5: This is beyond the scope of this paper. But is it possible to extend WorldSense to an open-ended or generative benchmark? This would make it more aligned with user-end usage.**
>
> Thanks for your suggestions. As explained in ***our response to Q2***, we have clarified the reasons for using only multiple-choice questions, and Worldsense can be easily converted into open-ended questions. We will add open-ended questions in our future work to conduct a more comprehensive evaluation.

---

### Official Review · Reviewer_ZaHB · 2025-11-02

**Soundness:** 3
**Presentation:** 3
**Contribution:** 3
**Rating:** 6
**Confidence:** 3

**Summary:**

The paper introduces WorldSense, a new benchmark designed to evaluate how MLLMs understand and reason about real-world video-audio-text inputs. WorldSense integrates both audio and video to test true omni-modal understanding.
The dataset includes 1,662 synchronized videos across 8 domains and 67 categories, with 3,172 multiple-choice QA pairs spanning 26 different tasks. These questions require both audio and visual cues to answer correctly. The authors also evaluate various open-source and proprietary MLLMs, showing that even advanced models like Gemini 2.5 Pro only achieve 65.1% accuracy, exposing significant gaps in real-world multimodal reasoning.

**Strengths:**

- A novel audio-video benchmark focused on omni-modal understanding.
- Expert curated by annotators and manual QA design for better quality.
- Paper is well-written.

**Weaknesses:**

- Although, the dataset is diverse, the distribution of question difficulties across categories or cognitive levels is not very clear. Some tasks might be more perception-heavy than reasoning-heavy, which could bias model comparison.

**Questions:**

- How do you ensure that questions cannot be answered from text transcripts alone?
- Will the benchmark be publicly released, and under what license?

---

> ### Author Response · Authors · 2025-11-20
> **Response to Reviewer ZaHB**
>
> We sincerely appreciate your recognition of the contribution of our benchmark. We believe the following responses would resolve all concerns.
>
>
> > **Q1: Although, the dataset is diverse, the distribution of question difficulties across categories or cognitive levels is not very clear. Some tasks might be more perception-heavy than reasoning-heavy, which could bias model comparison.**
>
> Thanks for your suggestion. In fact, WorldSense is relatively balanced in terms of perception and reasoning tasks, and we will clarify this from the following two perspectives:
>
> 1. **Balanced Category.** ***Figure 2*** presents the number and distribution of questions across perception and reasoning tasks, showing a relatively balanced distribution among question and reasoning. Additionally, we provide a performance comparison of each model across different and task types in ***Figure 4***.
>
>
> 2. **Performance across Category.**  ***Table 2*** further displays model performance across various video categories. It can be observed that the performance of the same model across different video categories is quite similar, indicating a well-balanced distribution of our video categories. Furthermore, ***Figure 4*** reveals that model performance gradually declines with increasing task difficulty: accuracy is highest for perception-related questions and lowest for reasoning-related questions, which aligns with the expected performance patterns of existing models. These experimental results demonstrate the limitations of existing models in omni-modality understanding, while also validating the value of WorldSense.
>
>
> Worldsense achieves **a balanced distribution** both in video categories and task types, and we provide fine-grained performance data that allows other researchers to easily conduct in-depth comparisons and analyses of model performance. We will emphasize this point in the revised manuscript.
>
>
>
> > **Q2: How do you ensure that questions cannot be answered from text transcripts alone?**
>
> As described in ***Section 3.2***, our data annotation pipeline can effectively ensure that the questions cannot be answered solely by text transcripts. First, during the annotation process, we require annotators to annotate questions by fully integrating video and audio information. Second, we conduct manual reviews to exclude questions that can be answered solely using text transcripts. Through these two methods, we can effectively ensure that the questions cannot be answered from text transcripts alone.
>
>
> > **Q3: Will the benchmark be publicly released, and under what license?**
>
> We will release the benchmark once accepted.

---

### Comment · Area_Chair_GpdL · 2025-11-25
**Discussion Period**

Dear Reviewers and Authors,

Thank you to the authors for submitting your rebuttal. We kindly encourage reviewers to take a moment to read the response and share any follow-up thoughts. Your timely engagement at this stage is highly valuable and helps ensure a fair, well-informed final decision.

We appreciate everyone’s efforts and contributions to the process.

Warm regards,

---

### Author Response · Authors · 2025-12-02
**Summary from the Authors**

Dear  Area Chair:

We sincerely appreciate the constructive feedback provided by all reviewers and the time dedicated to evaluating our work on WorldSense.

> **Summary of Discussion**

During the discussion period, we engaged with the reviewers to address their concerns regarding task definitions, model comparisons, and experimental details. We are pleased to note that the feedback has been largely positive, with reviewers acknowledging the value and novelty of our benchmark.

- ***Reviewer zBjg (Nov 27, 02:40 AOE)*** maintains the score of 6, and claim "Thanks for the rebuttal... I will keep my rating and tend to accept this paper."

- ***Reviewer E5me (Nov 26, 14:24 AOE)*** increases the score from **4 to 6**. The reviewer acknowledged our response regarding the benchmarks comparison and the inclusion of recent powerful models.

- ***Reviewer Fj9V*** maintains a strong score of **10**. The reviewer highlights the work as a "Strong accept" and suggests it "should be highlighted at the conference as spotlight or oral."

- ***Reviewer ZaHB*** maintains the score of 6. Acknowledged the contribution as "A novel audio-video benchmark focused on omni-modal understanding."



> **Summary of Rebuttal**

First, we are encouraged to note that the reviewers provided highly positive assessments of our contributions:

- **Motivation:** The benchmark addresses the critical need for "omnimodal reasoning," distinguishing itself from existing works by requiring joint reasoning over synchronized visual and audio inputs (Reviewers Fj9V, ZaHB).
- **Data Quality:** The "expert curated" annotations and manual QA design are recognized as a guarantee for quality, which is "very rare these days" (Reviewers ZaHB, Fj9V).
- **Comprehensive Evaluation:** The experiments covering open-source and proprietary MLLMs provide a "benchmarking foundation for future research" (Reviewer Fj9V) and are deemed "sufficient and convincing" (Reviewer E5me).

Reviewers raised several insightful questions. We conducted additional analyses and clarifications, which can be categorized into three main areas:

1. **Regarding Benchmark Design and Methodology:**

- **Omni-modal Definition:** We clarified that WorldSense evaluates the synergy of Audio and Video. While we use "omni-modal" to describe this holistic understanding, we clarified that captions are used primarily for ablation. We address concerns about the term by emphasizing the strict requirement for cross-modal integration (Response to Fj9V, zBjg).
- **Task Balance:** We demonstrated that WorldSense is balanced between perception and reasoning tasks. We provided data showing that model performance declines as task difficulty increases (reasoning tasks), validating the benchmark's stratified difficulty (Response to ZaHB, Fj9V).
- **QA Format:** We justified the use of Multiple-Choice QAs for standardized evaluation (following Video-MME) and ease of assessment, while noting that the dataset can be readily adapted to open-ended formats (Response to Fj9V, zBjg).
- **Text Shortcuts:** We reiterated our strict annotation pipeline and manual reviews to ensure questions cannot be answered via text transcripts alone (Response to ZaHB).

2. **Regarding Comparisons and Related Work:**

- **Comparison with Existing Benchmarks:** We provided a detailed comparison against AVUT and DailyOmni. We highlighted that WorldSense is significantly larger (3,172 QA pairs vs. ~1k), covers broader domains (8 domains vs. audio-centric), and includes deeper reasoning tasks compared to these concurrent works (Response to E5me).

3. **Regarding Experiments and Analysis:**

- **New Model Baselines:** In response to Reviewer E5me, we expanded our evaluation to include recent powerful models (e.g., Qwen3-Omni, Video-SALMONN 2). The results confirm that even the latest models struggle on WorldSense, reinforcing the benchmark's challenge and longevity.
- **Failure Analysis:** We added a detailed error analysis (Section 4.3) categorizing failure modes into Inadequate Audio Understanding, Limited Cross-Modal Integration, and Insufficient Complex Reasoning, providing deeper community insights (Response to zBjg).

Once again, we extend our heartfelt thanks to all reviewers and the ACs for their diligent efforts and significant contributions to improving WorldSense.

Best regards,

The Authors

---

### Meta-Review · Area_Chair_ZxiP · 2025-12-26

**Summary:**

In the first round of review, reviewers confirmed the novelty, thorough experiments and presentation of this paper. During the discussion period, the authors solved the concerns regarding task definitions, comparisons with existing approaches and experimental details. After discussion, all reviewers approved the contributions of this study and the responses provided by authors. I thereby recommend to ACCEPT this paper.

**Reviewer Concerns:**

Regarding the benchmark, the reviewers raised concerns about Omni-modal definition and task balance. For evaluation the authors suggested more comparison with existing benchmarks and failure case analysis. The responses show that all these concerns were responded/solved by the authors after rebuttal.

**Reviewer Scores:**

The initial scores are 10, 6, 6,4 where 4 was changed to 6 (and reversed) after discussion. I do believe the 4 score will be changed to 6.

---

### Decision · Program_Chairs · 2026-01-26

Accept (Poster)